# Establishment of a Two-Stage Turbocharging System Model and Analysis on Influence Rules of Key Parameters

**Wei Tian [1], Defeng Du [1], Juntong Li [2], Zhiqiang Han [2] and Wenbin Yu [3],***

[1]  Vehicle Measurement, Control and Safety Key Laboratory of Sichuan Province, Xihua University, Chengdu 610039, China; tianviv@163.com (W.T.); dudefeng0918@foxmail.com (D.D.)

[2]  Fluid and Power Machinery Key Laboratory of Ministry of Education, Xihua University, Chengdu 610039, China; Lijuntong243@163.com (J.L.); hanzq@mail.xhu.edu.cn (Z.H.)

[3]  Department of Mechanical Engineering, National University of Singapore, 9 Engineering Drive 1, Singapore 117576, Singapore

*  Correspondence: wbyu@u.nus.edu; Tel.: +65-84375901

**Abstract:** This paper took a two-stage turbocharged heavy-duty six-cylinder diesel engine as the research object and established a two-stage turbocharging system matching model. The influence rules between the two-stage turbocharging key parameters were analyzed, while summarizing an optimization method of key parameters of a two-stage turbocharger. The constraint equations for the optimal distribution principle of the two-stage turbocharger's pressure ratio and expansion ratio were proposed. The results show that when the pressure ratio constraint equation and expansion ratio constraint equation are satisfied, the diesel engine can achieve the target pressure ratio, while the total energy consumption of the turbocharger is the lowest.

**Keywords:** two-stage turbocharger; pressure ratio; expansion ratio; matching model constraint equation

## 1. Introduction

Two-stage turbocharging technology has attracted much attention at home and abroad as a highly promising turbocharging technology. Exhaust gas recirculation, homogeneous combustion, Miller cycle, etc., [1] have been applied to turbocharging technology. However, with increasingly strict regulations on hazardous emissions, the two-stage turbocharging system needs to be further improved. A number of model descriptions have been proposed for the matching of the two-stage turbocharging system. Benson and Svetnicka et al. developed a mathematical model, predicting the matching characteristics of a two-stage turbocharged diesel engine [2]. The test results show that the predicted diesel engine operating point is quite close to that acquired from the test, and the method provides a calculation procedure to accurately predict the external air intake required for the two-stage turbocharged diesel engine under various working conditions. Bahiuddin et al. [3] and Chiong et al. [4] predicted the performance parameters of a turbocharger based on different instantaneous isentropic efficiencies by the VGT (Variable Geometry Turbocharger) control method. The model can reach the experimental matching target value with low computational amount and improved accuracy under steady state and transient conditions. Tancrez et al. [5] modeled the compressor and turbine to match the engine and analyzed the relationship between the compressor efficiency and the pressure ratio, and that between the mass flow and the pressure ratio when the turbocharger mass flow was 0.01–0.08 kg/s and the efficiency was 0.25–0.65. The results show that as the pressure ratio increases, the mass flow rate increases, when the expansion ratio is greater than 2.5, the mass flow rate tends to be gentle, when the pressure ratio is greater than 1.5, the compressor efficiency increases, and the pressure ratio

increases accordingly. Galindo [6], Zheng [7], and Yang et al. [8] established an analytical model of the function of the parameters of the engine and turbocharging system based on the relationship between the pressure ratio and the expansion ratio. The results indicate that the distribution ratio of the pressure ratio, total turbine expansion ratio, and equivalent efficiency are the main factors affecting turbocharge pressure and engine performance. Liu et al. [9] and Sanaye et al. [10] also considered the influence of two-stage turbocharging distribution and turbine flow on the basis of the former and established a turbocharger calculation model and genetic algorithm (GA) optimization technology. Compared with the conventional methods, the matching method enables the matching points under different operating conditions to be located in a more reasonable area on the compressor map Li [11], Fenghua [12], and Shi et al. [13] carried out theoretical research on the matching problem of an adjustable two-stage turbocharging system at different altitudes. Through the theoretical analysis of the adjustable two-stage turbocharging system, the equivalent matching model of the turbocharging system and the diesel engine is established, which provides abundant data for studying the matching model of a turbocharged engine at different altitudes. Besides, De Bellis et al. [14] established a turbocharging model with wastegates, predicted the mass flow at turbine inlets under different operating conditions based on the GT-power turbine model, evaluated the steady-state flow performance of the turbine, and made verification through the test. The model matching method is more accurate and faster than the traditional methods. Bo et al. of Shanghai Jiaotong University proposed a matching method for an adjustable two-stage turbocharging system of a diesel engine based on adjustment capability, and provided the matching criterion of an adjustable two-stage turbocharging system from the economic point of view according to the equivalent turbocharger concept, i.e., exhaust energy distribution is compatible with efficiency to maximize the efficiency of the turbocharging system [15]. However, the above research has not established a model from the perspective of energy distribution optimization to propose the theoretical equation for the optimal distribution principle of the pressure ratio and the expansion ratio of a two-stage turbocharger.

Many scholars have conducted in-depth research on the interaction between key parameters of turbocharging systems. Manivannan et al. [16] experimentally proved that the turbocharger inlet air temperature is determined by the turbine expansion ratio. If the effect of the expansion ratio on the temperature is not taken into account, the pressure drop inside the turbine may cause the air condensation, thereby affecting the test temperature results. With the use of the mathematical calculations of the NAPL (Nonaqueous Phase Liquid) boundary, the energy conservation equation and the Meanline integral method, Saipom et al. [17] processed the test results through a series of experiments, finding that the efficiency of the turbocharger is jointly determined by the measured inlet/outlet temperature, pressure and flow of the turbine, and those on both sides. Studies have shown that the higher the inlet temperature is, the higher the efficiency of the turbocharger is. Through one-dimensional performance analysis of a turbocharger turbine, Ding et al. [18] obtained different expansion ratios based on tests and evaluated turbine performance based on the inlet conditions at each moment. The method can effectively avoid the influence of other parameters on the test and help more accurately and quickly understand the turbine performance. To improve the efficiency of the turbine, Zhao et al. [19] have shown through experiments that higher efficiency can be achieved by appropriately increasing the expansion ratio and the pressure ratio, but the coupling relationship between the expansion ratio and the pressure ratio is not considered in this test. To continuously improve the efficiency of the turbocharger, Bo et al. [20,21] conducted a numerical analysis of the turbocharger flow through a CFD (Computational Fluid Dynamics) simulation and experimental measurements and analyzed the relationship between the two, finding that the improvement of the compressor inlet air flow quality can effectively improve the efficiency of the turbocharger. In addition, Sanaye et al. [10], Shi et al. [19] and Lei et al. [22] tested the different inlet pressures of the turbocharger. The study shows that when the turbocharger is expanded to the required pressure by the ambient pressure, the air density and efficiency can be increased. However, none of the above researchers have

simultaneously considered the relationship between the efficiency of the low-pressure/high-pressure stage turbocharger and total efficiency.

For the turbocharged system matching model, domestic and foreign scholars have proposed different constraints. Jiao et al. [20,22,23] used the turbocharger numerical simulation and the MP (Maximum Parsimony) method to specify the inlet pressure with atmospheric pressure as the boundary condition, so that each fluid region of the turbine is regarded as a steady state with the spatial averaging of the smooth data of adjacent regions on the mixing plane. The results show that this method eliminates any instability caused by the circumferential variation of the channel flow field, resulting in steady-state results. Padzillah et al. [24] conducted tests for the turbocharger under different pulsating flow conditions at 30,000 rpm of the turbine and with the outlet static pressure at the outlet boundary to correct the coefficient modification during steady-state operation of the turbocharger. The results show that the method can effectively reduce error and the error of the mass flow parameter of 8% can be reduced at best. In addition, Aymanns et al. [25] studied the turbocharger and found that the closer the pressure loss boundary position is to the turbine, the more stable the pressure is, and the more stable the expansion ratio and the pressure ratio is. Chiong et al. [4] proposed a pressure boundary hypothesis using the traditional mean line integration method and the physical extrapolation method, verifying whether the mass flow rate and turbocharger efficiency are good and predicting whether the engine performance meets the requirements through the pressure steady-state simulation. The research results show that the results from the method are consistent with the previous research results, and the calculation is simpler and faster.

The above literature shows that many scholars have conducted in-depth research on the relationship between turbocharging matching models and key parameters. However, there is no systematic and in-depth study on the influence of key gas circuit parameters, such as efficiency of high-pressure stage turbocharger, efficiency of low-pressure stage turbocharger, and front–rear temperature difference of high-pressure stage turbine on the two-stage turbocharge matching relationship. By deducing the relational expression between the two-stage turbocharged system compressor and the turbine energy balance, the authors have concluded an optimization method for the two-stage turbocharger's key parameters, proposed the theoretical constraint equations for optimal distribution principle of pressure ratio and expansion ratio of a two-stage turbocharger, and analyzed the influence rules of key gas circuit parameters of the two-stage turbocharged diesel engine, providing a theoretical basis and data support for the rational study of the two-stage turbocharging system matching model.

## 2. Establishment of Two-Stage Turbocharging System Model

### 2.1. Model Hypotheses

To analyze the influence of key parameters of the two-stage turbocharging system, the authors established a thermodynamic model and proposed the following assumptions:

① The working medium in the cylinder is in a uniform state, that is, the pressure, temperature, and concentration are equal at each point at the same instant in the cylinder, ② the gas remaining in the cylinder and the gas flowing into the cylinder are instantaneously and completely mixed during the intake period, ③ the surge and blocking areas of the turbocharger are not considered, and ④ the in-cylinder working medium has no leakage in the closing process [26,27].

### 2.2. Model Equations

The model logic diagram is shown in Figure 1.

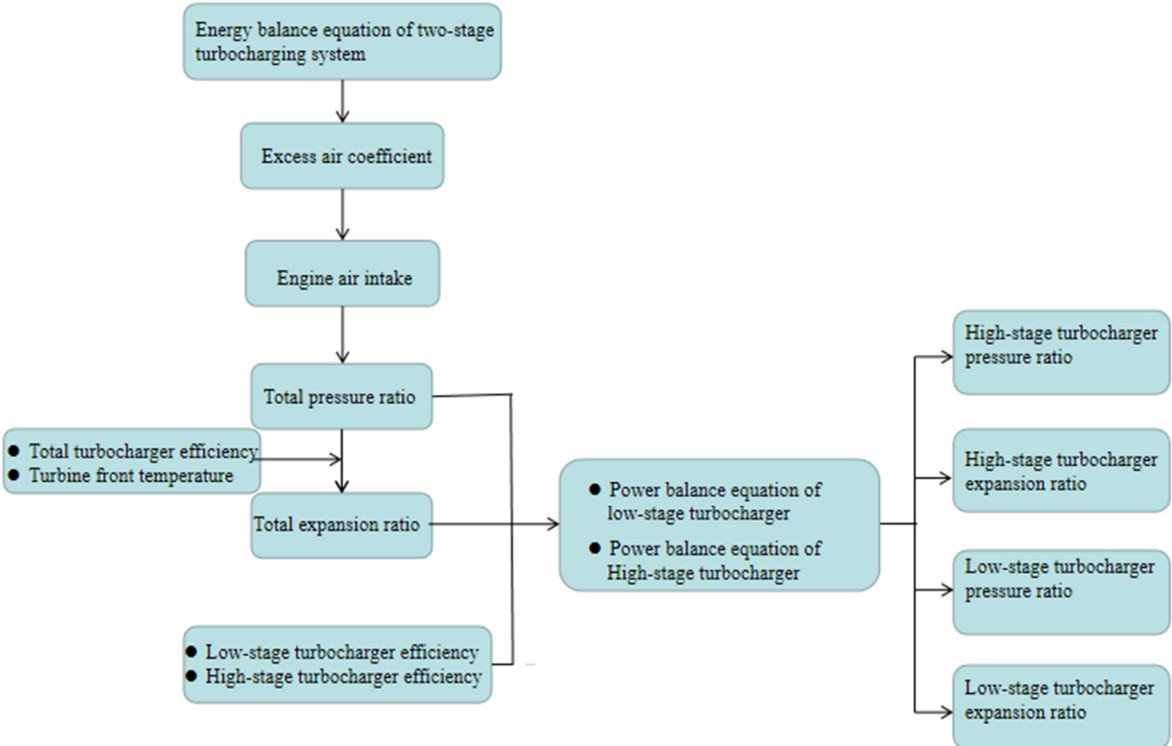

**Figure 1.** Calculation model logic diagram.

According to the gas parameters flowing out and flowing into the cylinder of the diesel engine, the energy balance is established, and the energy balance equation is:

$$\xi_T H_u - \frac{3600}{b_e \eta_m} + \phi_{as} a (uc_p)_a T_a = (\phi_{as} - 1 + \beta_0) a (uc_p)_T T_T \tag{1}$$

The coefficient of heat utilization is expressed as:

$$\xi_T = 1.028 - 0.00096 \frac{(\phi_a b_i)^{0.5} T_S^{1.6} V_m^{0.78} (0.5 + D/2S)}{(Dn) \cdot (DP_S)^{0.22}} \tag{2}$$

The molar heat capacity at constant pressure at the turbine inlet and outlet is respectively expressed as:

$$(uc_p)_T = 8.315 + \frac{20.47 + (\phi_{as} - 1) \cdot 19.26}{\phi_{as}} + \frac{3.6 + (\phi_{as} - 1) \cdot 2.51}{1000 \phi_{as}} \tag{3}$$

$$(uc_p)_a = 27.59 + 0.0025 T_a \tag{4}$$

The indicated specific fuel consumption of the engine is:

$$b_i = b_e \eta_m \tag{5}$$

The mechanical efficiency is:

$$\eta_m = \frac{P_{me}}{P_{me} + D^{-0.2} (0.00855 V_m + 0.0789 P_{me} - 0.0214)} \tag{6}$$

The compressor flow is:

$$q_{mb} = \frac{14.3 P_e b_e \phi_{as}}{3600} \tag{7}$$

The turbine flow is:

$$q_{mT} = 1.015 \times q_{mb} \tag{8}$$

The air pressure of the intake pipe in the engine cylinder is:

$$P_S = \frac{(\phi_a \cdot \phi_s) \cdot P_{me} T_S b_e}{894.7(\phi_c \phi_a)} \tag{9}$$

The turbocharge pressure is:

$$P_b = P_S + (0.03 \sim 0.05) \tag{10}$$

From the above formula, the excess air ratio $\phi_a$, total excess air ratio $\phi_{as}$, turbocharge pressure $P_b$, and compressor flow $q_{mb}$ are obtained.

Considering the two-stage turbocharging system as a single-stage turbocharging system, the relationship between the turbine front pressure, $P_T$, and the total efficiency of the turbocharger, $\eta_{Tb}$, is obtained from the power balance relational Expression (11) between the turbine and the compressor.

$$\eta_{Tb} \frac{q_{mT}}{75} \frac{\kappa_T}{\kappa_T - 1} R_T T_T (1 - \frac{1}{(\frac{P_T}{P_{T_0}})^{\frac{\kappa_T - 1}{\kappa_T}}}) = \frac{q_{mb}}{75} \frac{\kappa}{\kappa - 1} RT_0 [(\frac{P_b}{P_0})^{\frac{\kappa - 1}{\kappa}} - 1] \tag{11}$$

### 2.3. Relationship between Turbine Front Pressure $P_T$ and Total Turbocharger Efficiency $\eta_{Tb}$

Based on the external characteristic boundary condition parameters of a WP12 heavy-duty diesel engine, the relationship between the turbine front pressure, $P_T$, and the total turbocharger efficiency, $\eta_{Tb}$, is analyzed. The main technical parameters of the engine are shown in Table 1, the boundary conditions are shown in Table 2, and the calculation results are shown in Figure 1.

**Table 1.** Technical Parameters of a WP12 Heavy-duty Diesel Engine.

| Item | Parameter |
|---|---|
| Cylinder diameter (mm) | 126 |
| Stroke (mm) | 155 |
| Displacement (L) | 11.596 |
| Compression ratio | 17:1 |
| Intake swirl ratio | 1.2 |
| Combustion chamber | ω type |
| Intake type | Charging with inter-cooling |
| Number of holes × aperture × cone angle | 8 × 0.217 × 143 |
| Max. rotation speed (rpm) | 2200 |
| Max. power (kW) | 353 (2100 rpm) |
| Max. torque (Nm) | 1970 (1200–1500 rpm) |
| Max. explosion pressure (MPa) | 16.5 |

**Table 2.** Boundary Conditions.

| Input Parameter | 1300 rpm—100% Load | 1600 rpm—100% Load | 1900 rpm—100% Load |
|---|---|---|---|
| $V_m$ (m/s) | 6.717 | 8.267 | 9.82 |
| $P_{me}$ (MPa) | 0.2165 | 0.203 | 0.183 |
| $b_e$ (kg/kW·h) | 0.198 | 0.221 | 0.232 |
| $T_s$ (K) | 330 | 330 | 340 |
| $P_e$ (kW) | 271.97 | 313.96 | 336.43 |
| $T_T$ (K) | 811 | 879 | 973 |
| $P_{T0}$ (MPa) | 0.1065 | 0.11 | 0.113 |

$V_m$: piston mean speed (m/s), $P_{me}$: boost pressure (MPa), $b_e$: brake specific fuel consumption (g/(kW·h)), $T_s$: aero-engine inlet temperature (K), $P_e$: engine power rating (kW), $T_T$: temperature before turbine (K), $P_{T0}$: turbine outlet pressure (MPa).

As shown in Figure 2, at three different speeds of the diesel engine, as the total turbocharger efficiency, $\eta_{Tb}$, increases, the turbine front pressure decreases continuously. This is because the rise in the turbine front temperature, $T_T$, results in a decrease in the thermal efficiency of the diesel engine. When the total turbocharger efficiency, $\eta_{Tb}$, is 0.58, the turbine front pressures at three speeds are 0.37 MPa, 0.379 MPa, and 0.4015 MPa, respectively.

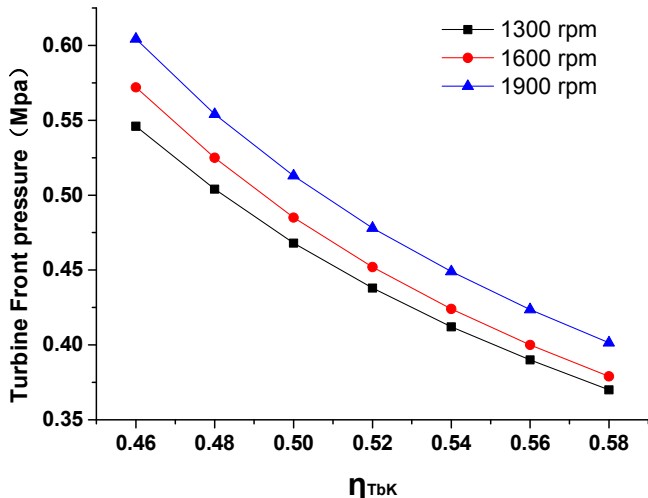

**Figure 2.** Relationship between the total turbocharger efficiency and the turbine front pressure.

### 2.4. Analysis of Key Parameters of Two-Stage Turbocharging System

There is also a power balance relationship between the turbines and the compressors of the two-stage turbocharging system. The power balance relationship between the high-pressure stage turbocharger turbine and the compressor is expressed as follows:

$$\eta_{Tbk}\frac{q_{mT}}{75}\frac{\kappa_T}{\kappa_T-1}R_T T_T\left(1-\frac{1}{\left(\frac{P_T}{P_{Th}}\right)^{\frac{\kappa_T-1}{\kappa_T}}}\right)=\frac{q_{mb}}{75}\frac{\kappa}{\kappa-1}RT_L\left[\left(\frac{P_b}{P_L}\right)^{\frac{\kappa-1}{\kappa}}-1\right] \tag{12}$$

The power balance relationship between the low-pressure stage turbine and the compressor is expressed as:

$$\eta_{TbL}\frac{q_{mT}}{75}\frac{\kappa_T}{\kappa_T-1}R_T T_{wL}\left(1-\frac{1}{\left(\frac{P_{Th}}{P_{T_0}}\right)^{\frac{\kappa_T-1}{\kappa_T}}}\right)=\frac{q_{mb}}{75}\frac{\kappa}{\kappa-1}RT_0\left[\left(\frac{P_L}{P_0}\right)^{\frac{\kappa-1}{\kappa}}-1\right] \tag{13}$$

It is known from Equations (12) and (13) that by adjusting the low-pressure stage efficiency, $\eta_{TbL}$, the high-pressure stage efficiency, $\eta_{Tbk}$, and the high-temperature stage turbine front-rear temperature difference, $T_T - T_{wL}$, the distribution ratio of the pressure ratio and the expansion ratio of the high-pressure/low-pressure stage turbocharger can be calculated.

2.4.1. Influence of Total Turbocharger Efficiency on Pressure Ratio and Expansion Ratio at Each Stage

In the case where the total pressure ratio and the total expansion ratio remain unchanged, the high-pressure stage turbocharger, $\eta_{Tak}$, is equal and the turbine front-rear temperature difference, $T_T - T_{WL}$, is 100 k, the pressure ratio of the low-pressure stage turbocharger is increased with the total turbocharger efficiency, as shown in Figure 3. Conversely, the pressure ratio of the high-pressure stage turbocharger decreases as the total turbocharger efficiency increases. Meanwhile, the expansion ratio of the low-pressure stage turbocharger decreases as the total turbocharger efficiency increases. Conversely, the expansion ratio of the high-pressure stage turbocharger increases as the total turbocharger efficiency

increases. This is because the total turbocharger efficiency is equal, that is, the total pressure ratio and the total expansion ratio are constant. Similarly, if the expansion ratio at the low-pressure stage is reduced, the expansion ratio of the high-pressure stage turbocharger will increase [28].

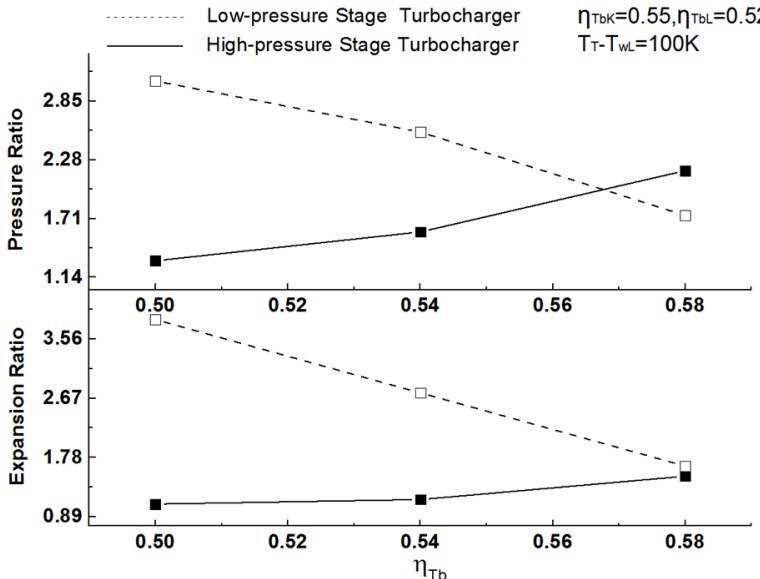

**Figure 3.** Relationship between the total turbocharger efficiency and the Pressure Ratio/Expansion Ratio.

### 2.4.2. Influence of Turbocharger Efficiency at Each Stage on Pressure Ratio and Expansion Ratio at Each Stage

When the total pressure ratio and the total expansion ratio are constant, the relationship between the efficiency of the low-pressure stage turbocharger and the two-stage pressure ratio/expansion ratio can be obtained by Formulas (12) and (13). As shown in Figure 4, when the diesel engine is at 1300 rpm speed, $\eta_{Tb} = 0.56$, high-pressure stage turbine front-rear temperature difference, $T_T - T_{WL}$, is 100 K, and the efficiency of the high-pressure stage turbocharger, $\eta_{Tbk}$, is controlled unchanged, if the low-pressure stage turbocharger efficiency, $\eta_{TbL}$, increases, the pressure ratio and expansion ratio of the low-pressure stage are increased while the turbocharge ratio and expansion ratio of the high-pressure stage are lowered. This is because the target intake air flow, the total turbocharge pressure, and the turbine front pressure are unchanged under the matched working conditions, so the total pressure ratio and the total expansion ratio are not changed and the total energy of the two-stage turbocharging system does not change. As $\eta_{TbL}$ increases and the low-pressure stage expansion ratio increases, the low-pressure stage energy distribution ratio increases, the low-pressure stage pressure ratio increases, and both the high-pressure stage pressure ratio and expansion ratio decrease, while when $\eta_{Tbk}$ remains unchanged, the high-pressure stage energy distribution ratio decreases.

Similarly, the relationship between the efficiency of the high-pressure turbocharger and the two-stage pressure ratio/expansion ratio can be obtained. As shown in Figure 5, with the increase in $\eta_{Tbk}$, the low-pressure stage pressure ratio and expansion ratio increase while the high-pressure stage turbocharge ratio and expansion ratio decrease.

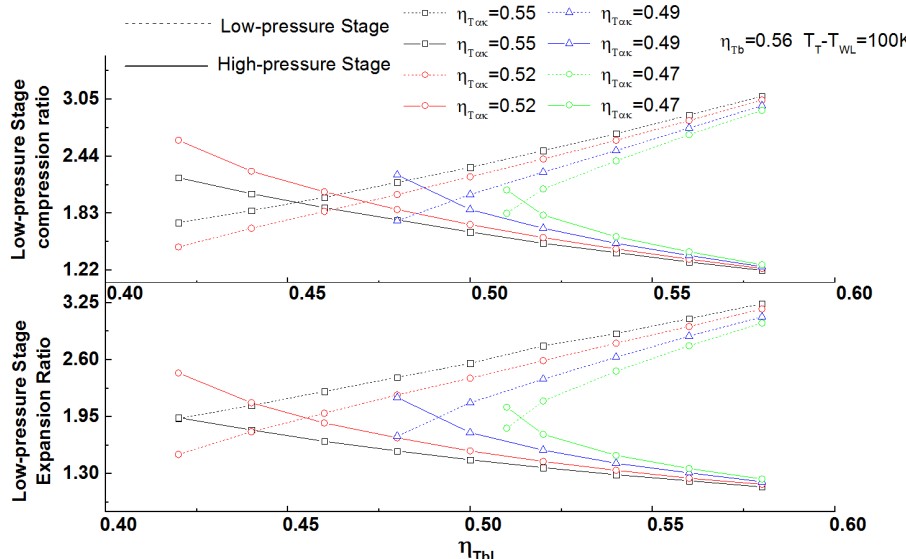

**Figure 4.** Effect of low-pressure stage turbocharger efficiency on two-stage pressure ratio and expansion Ratio.

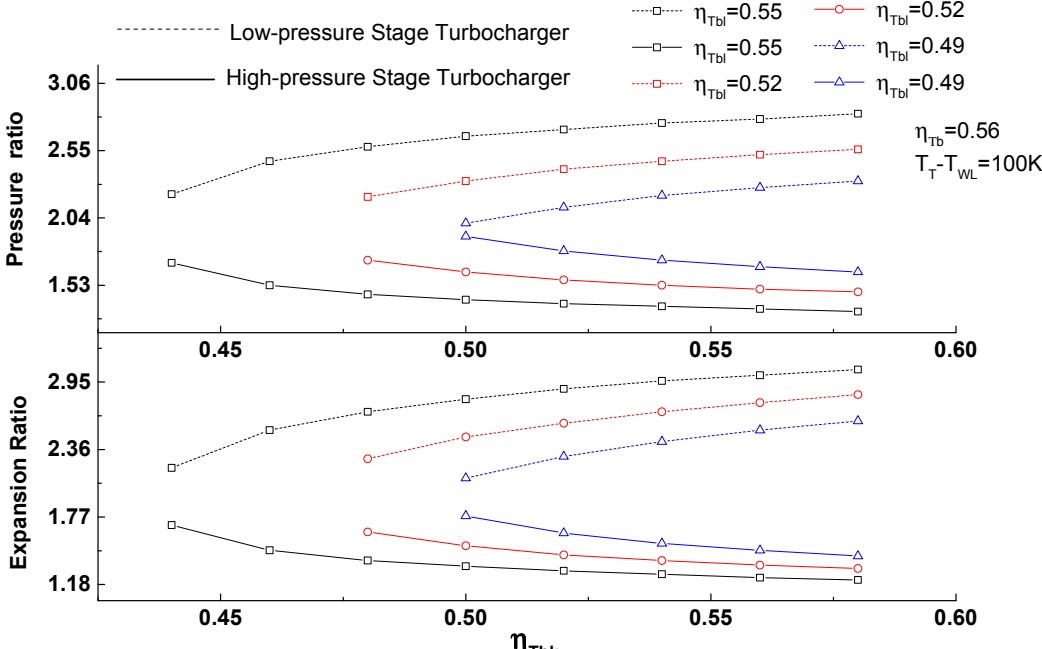

**Figure 5.** Effect of high-pressure stage turbocharger efficiency on two-stage pressure ratio and expansion ratio.

### 2.4.3. Influence of Front-Rear Temperature Difference of High-Pressure Stage Turbine on Pressure Ratio and Expansion Ratio at Each Stage

When the total pressure ratio and expansion ratio are constant and $\eta_{Tbk}$ and $\eta_{TbL}$ are kept unchanged at 1300 rpm and with $\eta_{Tb} = 0.56$, if the front–rear temperature difference of the high-pressure stage turbine $T_T - T_{WL}$ is increased, the energy consumed by the high-pressure stage turbine will become larger. This is because according to the law of conservation of energy, when $\eta_{Tbk}$ remains unchanged, the energy distribution ratio of the high-pressure stage becomes large, so both the pressure ratio and expansion ratio of the high-pressure stage increase with the increase of $T_T - T_{WL}$, and the

energy distribution ratio of the low-pressure stage is reduced, so both the pressure ratio and expansion ratio of the low-pressure stage decrease as $T_T$ - $T_{WL}$ increases, as shown in Figure 6.

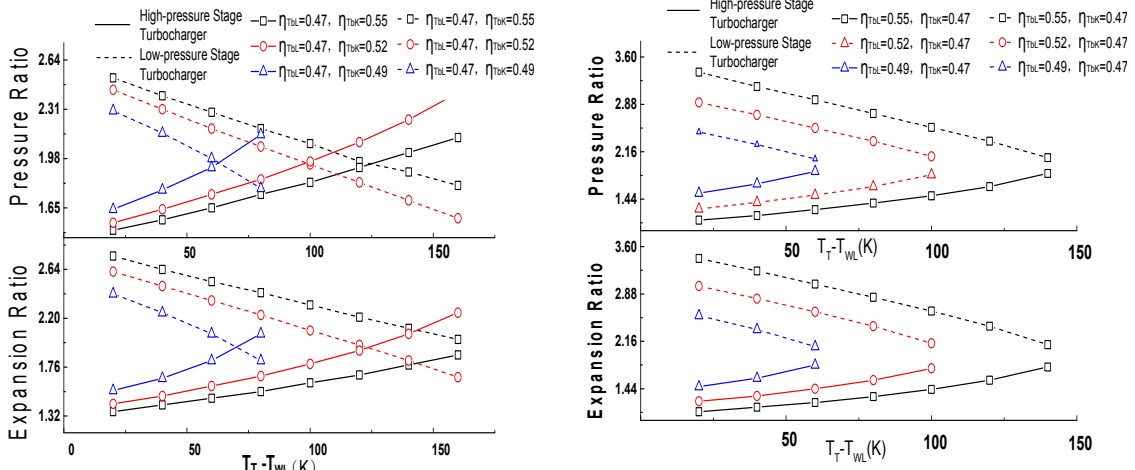

**Figure 6.** Effect of front–rear temperature difference of high-pressure stage turbocharger $T_T$ - $T_{WL}$ on pressure ratio and expansion ratio.

### 2.5. Optimization of Pressure Ratio Distribution Principle of Two-Stage Turbocharger

According to the above analysis, the energy distribution ratio of each stage in the two-stage turbocharging system can be realized by adjusting the high-pressure stage efficiency, $\eta_{T bk}$, the low-pressure stage efficiency, $\eta_{T bL}$, and the temperature drop of the high-pressure stage turbine $T_T$ - $T_{WL}$. In the two-stage turbocharging system, the distribution ratio of the pressure ratio of the high-pressure stage and the low-pressure stage directly affects the performance of the two-stage turbocharging system. Therefore, it is of significance to seek the distribution principle of the pressure ratio of the two-stage turbocharging system with the lowest energy consumption.

The total power of the compressor in the two-stage turbocharger is expressed as:

$$W_c = \frac{q_{mb}}{75}\frac{\kappa}{\kappa-1}RT_L[(\frac{P_b}{P_L})^{\frac{\kappa-1}{\kappa}} - 1] + \frac{q_{mb}}{75}\frac{\kappa}{\kappa-1}RT_0[(\frac{P_L}{P_0})^{\frac{\kappa-1}{\kappa}} - 1] \tag{14}$$

If the high-pressure stage intake air temperature, $T_L$, equals low-pressure stage intake air temperature, $T_0$ [29], through the intercooler control, the total power is expressed as:

$$W_c = \frac{q_{mb}}{75}\frac{\kappa}{\kappa-1}RT_0[(\frac{P_b}{P_L})^{\frac{\kappa-1}{\kappa}} + (\frac{P_L}{P_0})^{\frac{\kappa-1}{\kappa}} - 2] \geq \frac{2q_{mb}}{75}\frac{\kappa}{\kappa-1}RT_0\{[(\frac{P_b}{P_L})(\frac{P_L}{P_0})]^{\frac{\kappa-1}{2\kappa}} - 1\} \tag{15}$$

when $\frac{P_b}{P_L} = \frac{P_L}{P_0}$, the total power of the compressor in the two-stage turbocharger is the lowest.

Therefore, in order to achieve the target total pressure ratio, when the pressure ratio of the high-pressure stage turbocharger is equal to that of the low-pressure stage turbocharger, the total energy consumption of the two-stage compressor is the least.

### 2.6. Influence Analysis of Two-Stage Turbocharging Key Parameters with the Optimal Pressure Ratio Distribution of Two-Stage Turbocharger

2.6.1. Influence Relation between the Total Turbocharger Efficiency and the Efficiency of Turbocharger at Each Stage

Figure 7 shows the relationship between $\eta_{Tb}$, $\eta_{TbL}$, $\eta_{Tbk}$, and the two-stage expansion ratio with the diesel engine at a speed of 1300 rpm, the equal pressure ratio of the two-stage turbocharger, and the temperature of the high-pressure stage turbine decreased by 100 K. It can be seen from the figure that



when $\eta_{Tb}$ remains unchanged, $\eta_{Tbk}$ decreases with the increase of $\eta_{TbL}$. This is because the efficiency of the low-pressure stage turbocharger, $\eta_{TbL}$, increases when the total turbocharger efficiency, $\eta_{Tb}$, is constant. To maintain the energy balance of the two-stage turbocharging system, the low-pressure stage expansion ratio will decrease. Similarly, if the high-pressure stage turbocharger efficiency, $\eta_{Tbk}$, increases, the high-pressure stage expansion ratio is also reduced to maintain the balance of the energy distribution of the turbocharging system [30].

It can also be concluded from Figure 7 that when $\eta_{TbL}$ remains unchanged, as the total efficiency, $\eta_{Tb}$, increases, $\eta_{Tbk}$ gradually increases. This is because when the total turbocharger efficiency, $\eta_{Tb}$, is increased, the turbine front pressure, $P_T$, is lowered, and the total expansion ratio is reduced. Furthermore, because the low-pressure stage efficiency, $\eta_{TbL}$, is kept constant, the low-pressure stage expansion ratio is constant, and thus the high-pressure stage expansion ratio decreases accordingly. To reduce the energy balance of the two-stage turbocharging system, it is necessary to increase the efficiency of the high-pressure stage turbocharger, $\eta_{Tbk}$. As the high-pressure stage efficiency, $\eta_{Tbk}$, is maintained, the low-pressure stage efficiency, $\eta_{TbL}$, increases as the total efficiency, $\eta_{Tb}$, increases.

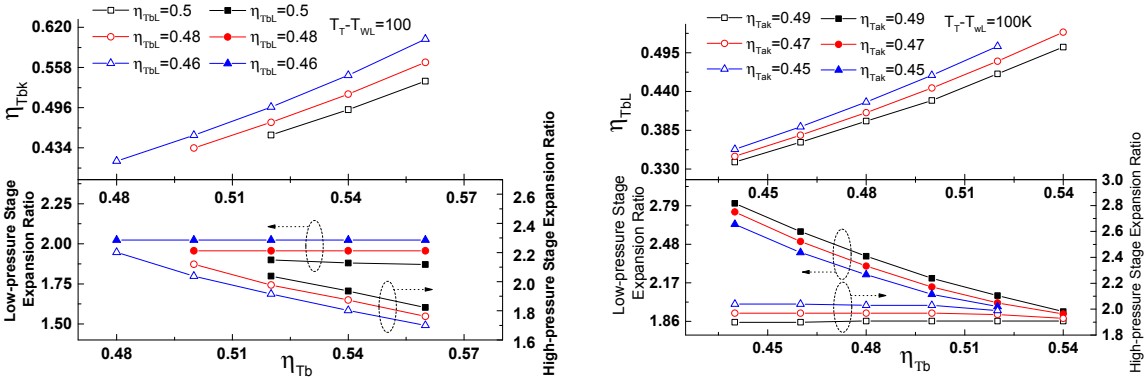

**Figure 7.** Relationship between the total turbocharger efficiency, the efficiency of the turbocharger at each stage, and the expansion ratio of each stage.

### 2.6.2. Relationship between Total Turbocharger Efficiency and Front–Rear Temperature Difference of High-Pressure Stage Turbine

It can be seen from Figure 8 that the efficiency of the low-pressure stage turbocharger decreases as the total turbocharger efficiency, $\eta_{Tb}$, increases when the high-pressure stage efficiency, $\eta_{Tbk}$, and the low-pressure stage efficiency, $\eta_{TbL}$, remain unchanged. This is because as the total turbocharger efficiency, $\eta_{Tb}$, increases, the turbine front pressure, $P_T$, decreases, resulting in a decrease in the total expansion ratio.

It can also be seen from Figure 8 that when the efficiency of the low-pressure stage turbocharger, $\eta_{TbL}$, and the efficiency of the high-pressure stage turbocharger, $\eta_{Tbk}$, are kept constant, the front–rear temperature difference of the high-pressure stage turbine $T_T - T_{WL}$ decreases as the total turbocharger efficiency, $\eta_{Tb}$, increases. The reason for this relationship is that as seen from Equations (12) and (13), when the high-pressure stage turbocharger efficiency, $\eta_{Tbk}$, and the low-pressure stage turbocharger efficiency, $\eta_{TbL}$, are constant, there is a wane and wax relationship between the high-pressure stage expansion ratio and the low-pressure stage turbine front temperature, $T_{wL}$, to maintain the energy distribution balance of the two-stage turbocharging system. That is, the change of expansion ratio of the low-pressure stage turbocharger is opposite to that of the high-pressure stage turbine front–rear temperature difference $T_T - T_{WL}$. Conversely, the change of expansion ratio of the high-pressure stage turbocharger is consistent with that of the high-pressure stage turbine front–rear temperature difference $T_T - T_{WL}$.

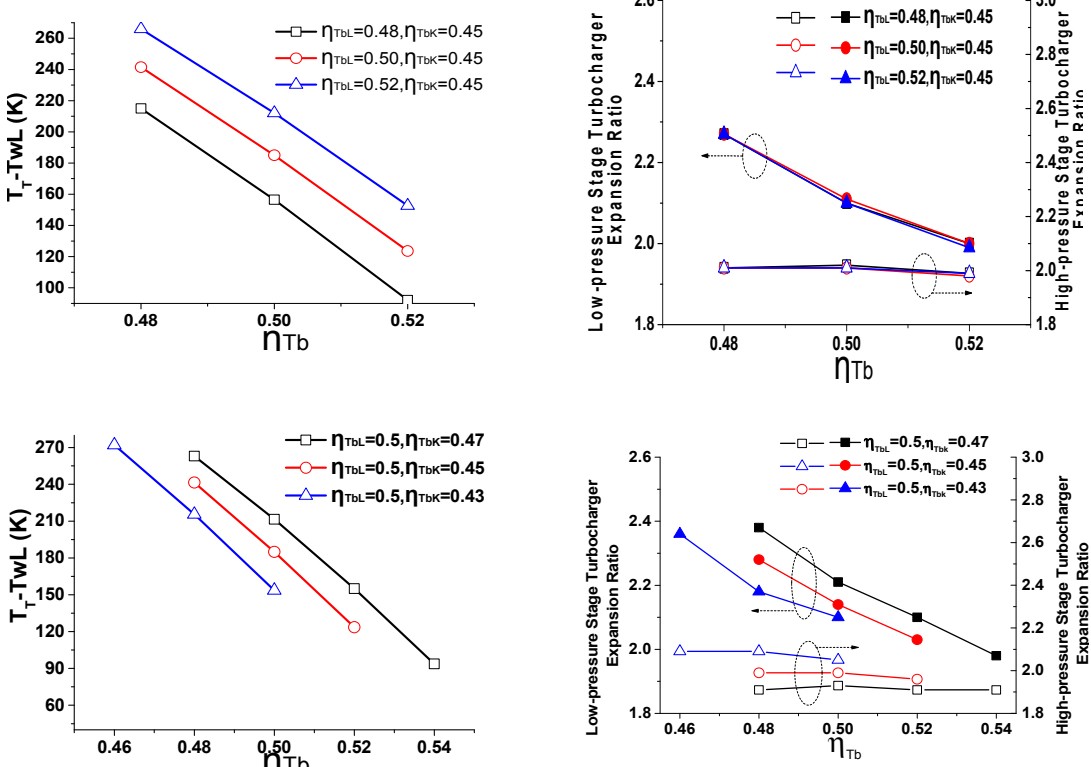

**Figure 8.** Relationship between the total turbocharger efficiency, the front–rear temperature difference of the high-pressure stage turbine $T_T$ - $T_{WL}$, and the expansion ratio at each stage.

### 2.7. Optimization of Expansion Ratio Distribution Principle of the Two-stage Turbocharger

After the two-stage pressure ratio is optimized to the lowest energy consumption, the expansion ratio can be optimized to realize the least energy consumption. The total power of the two-stage turbine can be known from the power balance Equations (12) and (13) of the turbine and compressor of the turbocharger at each stage:

$$\eta_{Tbk}\frac{q_{mT}}{75}\frac{\kappa_T}{\kappa_T-1}R_T T_T\left(1-\frac{1}{(P_T/P_{Th})^{(\kappa_T-1)/\kappa_T}}\right)+\eta_{TbL}\frac{q_{mT}}{75}\frac{\kappa_T}{\kappa_T-1}R_T T_{wL}\left(1-\frac{1}{(P_{Th}/P_{T_0})^{(\kappa_T-1)/\kappa_T}}\right) \quad (16)$$

When $(1-\frac{1}{(P_T/P_{Th})^{(\kappa_T-1)/\kappa_T}})\eta_{Tbk}T_T=(1-\frac{1}{(P_{Th}/P_{T_0})^{(\kappa_T-1)/\kappa_T}})\eta_{TbL}T_{wL}$, the two-stage consumes the least energy and the two-stage turbocharger can reach the target turbocharge ratio. If the expansion ratio of the high-pressure stage turbocharger and that of the low-pressure stage turbocharger are equal, Equation (16) can be simplified as:

$$\begin{aligned}&=\frac{q_{mT}}{75}\frac{\kappa_T}{\kappa_T-1}R_T\left(1-\frac{1}{(P_T/P_{Th})^{(\kappa_T-1)/\kappa_T}}\right)(\eta_{Tbk}T_T+\eta_{TbL}T_{wL})\\&\geq\frac{2q_{mT}}{75}\frac{\kappa_T}{\kappa_T-1}R_T\left(1-\frac{1}{(P_T/P_{Th})^{(\kappa_T-1)/\kappa_T}}\right)(\eta_{Tbk}T_T\eta_{TbL}T_{wL})^{0.5}\end{aligned} \quad (17)$$

When $\frac{\eta_{Tbk}}{\eta_{TbL}}=\frac{T_{wL}}{T_T}$, the target pressure ratio is reached. According to the calculation result of Equation (17), based on the empirical value of the front-rear temperature difference of the high-pressure stage turbine $T_T$ - $T_{WL}$ of the WP12 diesel engine under different working conditions, it can be concluded that the target pressure ratio is reached when $\eta_{TbL}$ is equal to (1.13–1.17) $\eta_{Tak}$, and the total energy consumption of the two-stage turbine is the least.

According to the above method, the model optimization calculation results are shown in Table 3 below.

**Table 3.** Results of key parameters of the turbocharger.

| Calculation Result | 1300 rpm—100% Load | 1600 rpm—100% Load | 1900 rpm—100% Load |
|---|---|---|---|
| $q_{mb}$ (kg/h) | 1458 | 1924 | 2129 |
| $P_b$ (MPa) | 0.376 | 0.435 | 0.447 |
| $P_T$ (MPa) | 0.379 | 0.446 | 0.449 |
| Low-pressure stage turbocharge ratio | 1.95 | 2.1 | 2.13 |
| High-pressure stage turbocharge ratio | 1.95 | 2.1 | 2.12 |
| Low-pressure stage expansion ratio | 1.9 | 2.05 | 2.06 |
| High-pressure stage expansion ratio | 1.87 | 1.97 | 1.98 |

*2.8. Calculation of Geometric Equivalent Circulation Cross Section of Two-Stage Turbine*

The effective equivalent circulation area of the turbine is calculated based on the following equations:

$$\mu_{TL}F_{TL} = \frac{q_{mT}}{\psi_{TL}\rho_{TL}\sqrt{2R_TT_{wL}}} \tag{18}$$

$$\mu_{TH}F_{TH} = \frac{q_{mT}}{\psi_{TH}\rho_{TH}\sqrt{2R_TT_T}} \tag{19}$$

where

$$\psi_{TL} = \sqrt{\frac{\kappa_T}{\kappa_T-1}[(\frac{P_{T_0}}{P_{Th}})^{2/\kappa_T} - (\frac{P_{T_0}}{P_{Th}})^{(\kappa_T+1)/\kappa_T}]} \tag{20}$$

$$\psi_{TH} = \sqrt{\frac{\kappa_T}{\kappa_T-1}[(\frac{P_{Th}}{P_T})^{2/\kappa_T} - (\frac{P_{Th}}{P_T})^{(\kappa_T+1)/\kappa_T}]} \tag{21}$$

and

$$\mu_T = 1.1787 - 1.6074/(20\pi_{T_{eq}} - 16) + 4.608e^{-18(\pi_{T_{eq}}-1.01)} \times (\pi_{T_{eq}} - 1.05) \tag{22}$$

The corrected expansion ratio is:

$$\pi_{T_{eq}} = \pi_T[1 - \frac{U_T^2(\kappa_T-1)}{\kappa_T R_T T_T}] \tag{23}$$

According to the above equations, the optimal geometry equivalent circulation area, $F_T$, of the turbine after optimization of the two-stage turbocharger can be calculated separately under each external characteristic working condition of the diesel engine. The specific calculation results are shown in Table 4 below. Considering that the realization of the high-density low-temperature combustion mode requires the speed external characteristic pressure ratio of 3.8 or more, and matching the turbocharger under full-load at medium- and high-speed will cause the diesel engine to fail to meet the design pressure demand under a low-speed working condition, the authors used the low-speed full-load point (maximum (max.) torque point) as the reference for the two-stage turbocharging system matching, and the compressor end and turbine end models of the final selected high/low-pressure stage turbocharger are shown in Table 5.

**Table 4.** Calculation results of turbine geometric equivalent circulation area.

| Calculation Result | 1300 rpm—100% Load | 1600 rpm—100% Load | 1900 rpm—100% Load |
|---|---|---|---|
| Geometric equivalent circulation area of the low-pressure stage turbine (cm$^2$) | 23.67 | 30.13 | 38.49 |
| Geometric equivalent circulation area of the high-pressure stage turbine (cm$^2$) | 13.79 | 16.43 | 19.65 |

**Table 5.** Model selection of the two-stage turbocharger.

| Turbocharger Provider | Turbocharger Location | Compressor End Model | Turbine End Model |
|---|---|---|---|
| Holset | Low-pressure stage turbocharger | B108N66RAH | HX50-S22W11 |
|  | High-pressure stage turbocharger | B85J71GL | C76N79FHB13UE |

## 3. Model Validation

*Test Platform and Test Device*

To verify the accuracy of the two-stage turbocharged engine's external characteristic gas circuit parameters calculated by the model matching method, the authors' research team built a 6-cylinder heavy-duty diesel engine test platform. The fuel injection system used the BOSCH second-generation high-pressure common rail system, the gas circuit system adopted two-stage turbocharged and dual EGR (Exhaust Gas Recirculation) (high-pressure EGR and low-pressure EGR) systems, and a bypass channel was respectively designed at the high-pressure stage turbine end and compressor end to adjust the operating state of the turbocharger and thus, to achieve the control of intake air status parameters. The separate or combined use of the high-pressure EGR system and the low-pressure EGR system can not only meet the fluctuation demands of the large-scale EGR rate in the test scheme, but also achieve the quick response of the diesel engine under transient operating conditions. Due to the need of the combustion scheme, the two-stage turbocharging system matched by the authors needs to achieve a pressure ratio of close to 4.0 under low-speed full-load conditions. Therefore, the test system was equipped with an IVCA (Intake Valve Late-closing) device to avoid the maximum pressure of the cylinder top dead center exceeding the limit (16.5 MPa), on the one hand. On the other hand, the Miller cycle is also achieved, and the power capacity is enhanced. The acquisition system consisted of a hot-film flowmeter, a transient fuel consumption meter, a temperature sensor, a pressure sensor, a photoelectric encoder, and an NI (National Instrument) data acquisition card. The data from the hot-film flowmeter and transient fuel consumption meter were transmitted to the acquisition card through 485 communication. The pressure sensor and the temperature sensor collected the steady-state pressure and temperature of the engine, the photoelectric encoder signal was collected to calculate the instantaneous engine speed, and the NI data acquisition card integrated all sensor data which were sent to the upper computer for processing, display, and storage. The test-bench schematic and test equipment installation diagram are shown in Figure 9 below. The diesel engine parameters and the detailed list of test equipment are respectively shown in Table 1, and in Table 6 below.

**Table 6.** Specifications and accuracy of measuring equipment.

| Equipment Name | Model | Specifications | Accuracy | Manufacturer |
|---|---|---|---|---|
| Thermal-film flowmeter | TOCEIL-20N125 | 0~1800 kg/h | 0.1 kg/h | Shanghai ToCeiL |
| Fuel consumption meter | CMFG010 | 0~60 Kg/h | ±0.12% | Shanghai ToCeiL |
| Electric eddy current dynamometer | CW440D | 440 kW 2800 N·m/1000–1500 rpm 6800 rpm | ±0.4% FS ±1 rpm | CAMA (Luoyang) |
| Pressure sensor | CYYZ11-Z-46-V5-07-B-G | −100–500 KPa | 0.1% FS | Beijing Star |
| Temperature sensor | WRNK-191 | 0–600 °C | 1% FS | Shanghai Yijia |
| Photoelectric encoder | E6C2-CWZ3E | 0~6000 rpm | 100 kHz | OMRON |
| Acquisition card | NI-USB6353 | −10 v~10 v | 16-bit resolution | National Instruments |

Figures 10–12 show the comparison between the experimental data and simulation results of the external characteristic of the two-stage turbocharged engine. The error is within 3.5%, which proves that the optimized matching method adopted by the two-stage turbocharger in this paper meets the target pressure ratio requirement.

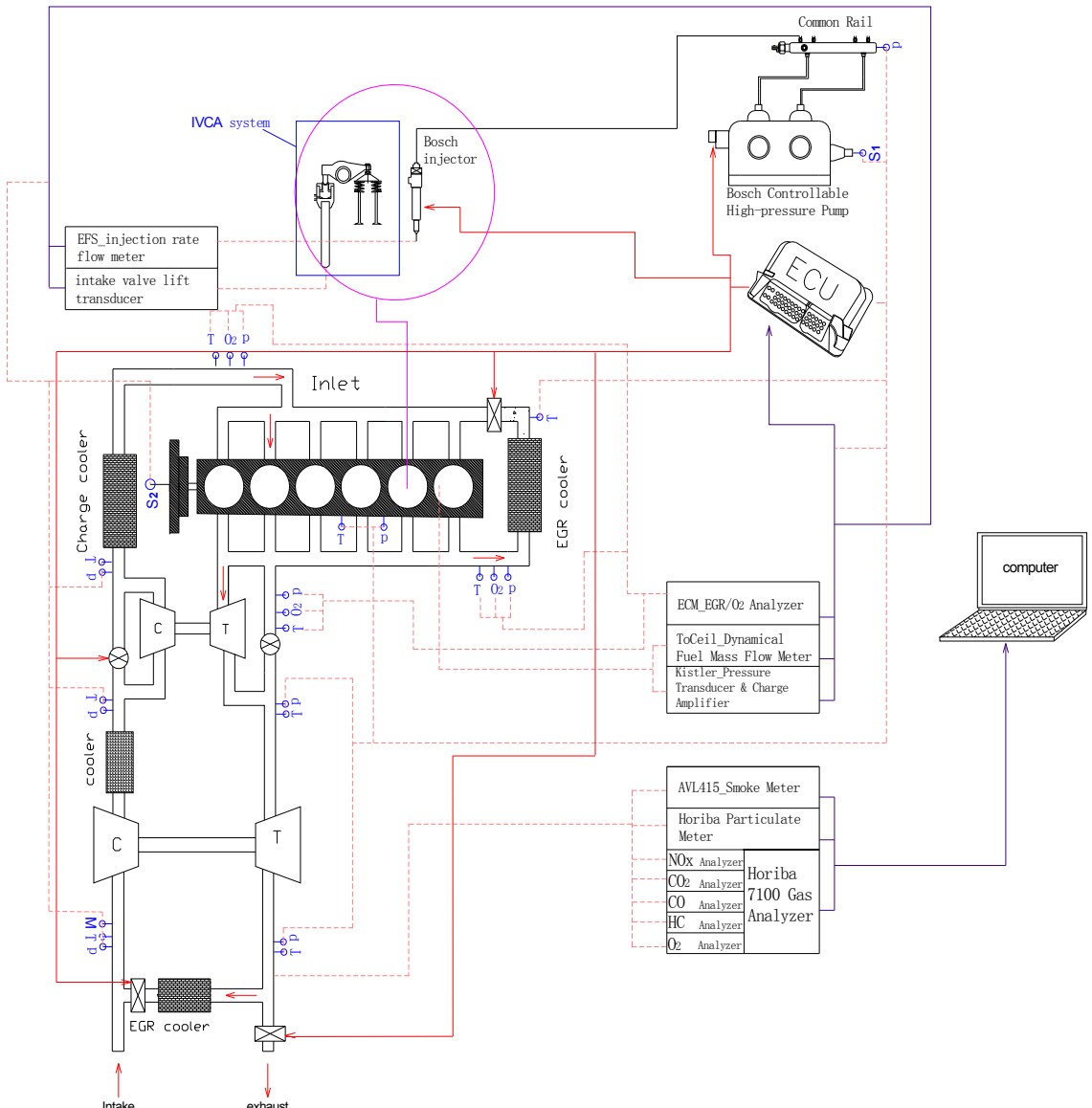

**Figure 9.** Schematic diagram of the test platform and test equipment. Note: P—pressure sensor, T—temperature sensor, M—intake flow sensor, S1—synchronization signal, S2—photoelectric encoder, $O_2$—oxygen sensor.

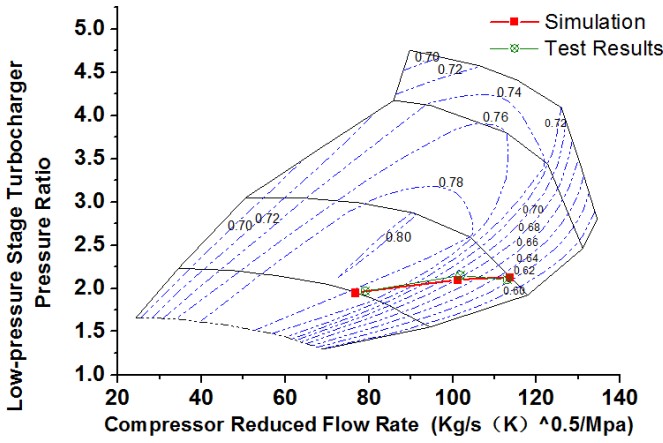

**Figure 10.** Comparison between low-pressure stage simulation and test results.

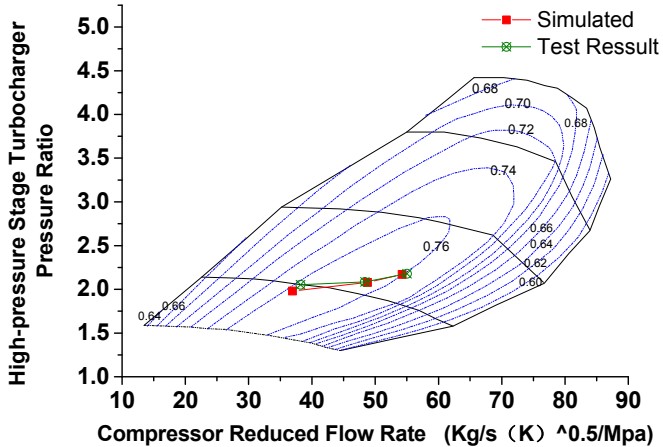

**Figure 11.** Comparison between high-pressure stage simulation and test results.

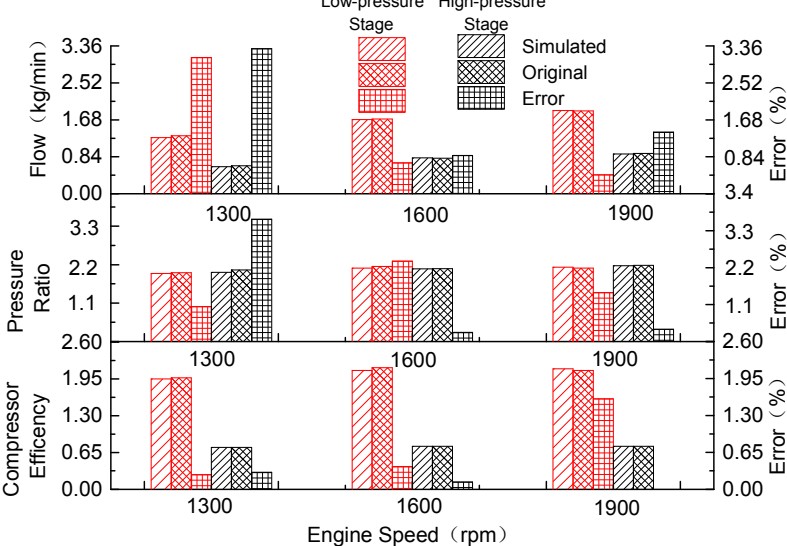

**Figure 12.** Comparison between results of low-pressure stage and high-pressure stage turbochargers.

## 4. Conclusions

This paper analyzed the influence rules of the key parameters of the two-stage turbocharging system through the power balance relationship between the turbine and the compressor and proposed the matching method for minimum energy consumption of the two-stage turbocharger as well as the theoretical constraint equations for the optimal distribution principle of pressure ratio and expansion ratio. The main conclusions are drawn as follows:

(1) According to the energy balance relationship between the compressor and the turbine of the two-stage turbocharging system, a method for optimizing the key parameters of the two-stage turbocharger was summarized, and the theoretical constraint equations for the optimal distribution principle of the pressure ratio and expansion ratio of the two-stage turbocharger was proposed, i.e.,

$$(1 - \frac{1}{(P_T/P_{Th})^{(\kappa_T-1)/\kappa_T}})\eta_{Tbk}T_T = (1 - \frac{1}{(P_{Th}/P_{T_0})^{(\kappa_T-1)/\kappa_T}})\eta_{TbL}T_{wL} \text{ and } \frac{P_b}{P_L} = \frac{P_L}{P_0}.$$

(2) Under the condition that the total pressure ratio, the total expansion ratio, the high-pressure turbocharger efficiency, and the front–rear temperature difference of the turbine are constant, the pressure ratio of the low-pressure stage turbocharger increases as the total turbocharger efficiency increases, while the pressure ratio of the high-pressure stage turbocharger decreases as the total turbocharger efficiency increases.

(3) When the total turbocharger efficiency, the high-pressure stage turbocharger efficiency, and the front-rear temperature difference of high-pressure stage turbine are constant, if the low-pressure stage turbocharger efficiency increases, the pressure ratio and expansion ratio of the low-pressure stage turbocharger increase and those of the high-pressure stage turbocharger are lowered.

(4) When the total pressure ratio, the total expansion ratio, and the efficiency of the turbocharger at each stage remain unchanged, both the pressure ratio and the expansion ratio of the high-pressure stage turbocharger increase with the increase of the front–rear temperature difference of the high-pressure stage turbine, while both the pressure ratio and the expansion ratio of the low-pressure stage turbocharger decrease as the front–rear temperature difference of the high-pressure stage turbine increases.

**Author Contributions:** The authors equally contributed to the deployment of the paper. Conceptualization, W.T. and Z.H.; methodology, Z.H.; software, J.L.; validation, W.T. and Z.H.; formal analysis, W.T., Z.H., and D.D.; resources, W.T. and Z.H.; data curation, J.L.; writing—original draft preparation, Z.H. and D.D.; writing—review and editing, W.T. and W.Y.; supervision, W.T. and Z.H.; project administration, W.T. and Z.H.; funding acquisition, W.T. and Z.H. All authors have read and agreed to the published version of the manuscript.

**Funding:** This research was financially supported by projects of the Award Program of the National Natural Science Foundation of China (51776177), the open project of the Key Laboratory for Vehicle Measurement, Control and Safety of Sichuan Province (SZJJ2013-028), and the open project of the Key Laboratory for Testing Fluids and Power Machinery of the Education Ministry of China (SZJJ2016-006).

**Conflicts of Interest:** The authors declare no conflict of interest.

## Abbreviations:

The following symbols are used in this paper:

| | |
|---|---|
| a | Theoretical air volume |
| $b_e$ | Brake specific fuel consumption (g/(kW·h)) |
| $b_i$ | Indicated specific fuel consumption (g/(kW·h)) |
| D | Piston diameter (mm) |
| $F_T$ | Geometric equivalent circulation area of turbine |
| $H_u$ | Calorific value of diesel (J/kg) |
| k | Air isentropic exponent |
| $k_T$ | Exhaust gas isentropic exponent |

| | |
|---|---|
| n | Engine speed (rpm) |
| $\zeta_T$ | Coefficient of heat utilization |
| $\phi_{as}$ | Total excess air ratio |
| $\phi_a$ | Excess air ratio |
| $\phi_c$ | Volumetric efficiency |
| $\phi_s$ | Scavenging coefficient |
| $\beta_0$ | Theoretical coefficient of molecular change |
| $T_a$ | Standard atmosphere pressure (K) |
| $T_T$ | Temperature ahead of turbine (K) |
| $T_S$ | Aero-engine inlet temperature (K) |
| $T_0$ | Compressor inlet temperature (K) |
| $P_S$ | Intake manifold pressure (MPa) |
| $P_T$ | Turbine inlet pressure (MPa) |
| $P_{T0}$ | Turbine outlet pressure (MPa) |
| $P_{Th}$ | Pressure ahead of low-pressure stage turbine (MPa) |
| $P_L$ | Outlet pressure of low-pressure stage compressor (MPa) |
| R | Air gas constant |
| $R_T$ | Exhaust gas constant |
| $T_{WL}$ | Temperature ahead of low-pressure stage turbine (K) |
| $q_{mb}$ | Compressor flow (kg/h) |
| $q_{mt}$ | Turbine flow (kg/h) |
| $\mu_T$ | Flow coefficient of the equivalent circulation cross-section of the turbine |
| $(\mu c_p)_T$ | Molar heat capacity at constant pressure at the average gas temperature of $T_T$ at the turbine inlet (J/(mol·K)) |
| $(\mu c_p)_a$ | Molar heat capacity at constant pressure at the air temperature of $T_0$ in the diesel intake pipe (J/(mol·K)) |
| $U_T$ | Turbine peripheral speed (m/s) |
| $V_m$ | Piston mean speed (m/s) |
| $\eta_{bL}$ | Low-pressure stage turbocharger efficiency |
| $\eta_{Tbk}$ | High-pressure stage turbocharger efficiency |
| $\eta_{Tb}$ | Total turbocharger efficiency |
| $\eta_m$ | Mechanical efficiency |
| $\psi_{TH}$ | Effective equivalent circulation area of high-pressure stage turbine (cm$^2$) |
| $\pi_T$ | Expansion ratio |
| $\pi_{T_{eq}}$ | Corrected expansion ratio |
| $\psi_{TL}$ | Effective equivalent circulation area of low-pressure stage turbine (cm$^2$) |
| $(1 - \frac{1}{(P_T/P_{Th})^{(\kappa_T-1)/\kappa_T}})\eta_{Tbk}T_T = (1 - \frac{1}{(P_{Th}/P_{T_0})^{(\kappa_T-1)/\kappa_T}})\eta_{TbL}T_{wL}$ | Expansion ratio constraint equation |
| $\frac{P_b}{P_L} = \frac{P_L}{P_0}$ | Pressure ratio constraint equation |

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
