# Peer review of "Establishment of a Two-Stage Turbocharging System Model and Analysis on Influence Rules of Key Parameters"

_energies, doi:10.3390/en13081953_

Round 1
Reviewer 1 Report
I have read carefully the manuscript entitled 'Establishment of Two-stage Supercharging System Model and Analysis on Influence Rules of Key Parameters', under consideration for publication in the Energies Journal.
The article considers the two-stage supercharging of a diesel engine. The authors present a model for the two-stage supercharging process, which they use to investigate the influence of key parameters on the system performance. The authors are then capable to derive constrains for optimal performance.
Despite the article describes an interesting approach to the problem at hand, there are many reasons that advise against the publication of this article, at least in its current form, in the Energies journal.
Therefore, I recommend the REJECTION of the article.
However, I encourage the authors to improve the points detailed below and and resubmit.
Major objections:
I proceed now to list several of the points that must be addressed before this work can be considered for publication in a peer-reviewed journal. The whole paper presents a sloppy appearence: it seems that an unifinished version of the manuscript has been submitted. Without being exhaustive, I find:
- The presentation of the “Model equations” is not informative enough. The authors simply enumerate the equations which are the base of their work, without any explanation of how these equations are derived or how of these equations interact with each other. As a results, following the paper becomes unnecessarily difficult. At this point, I believe that a diagram (perhaps something as simple as a block/flow diagram) relating the equations used throughout the paper with the facility considered would have been very helpful. In summary, in a posterior version of this article, the presentation of the equations needs to be made clear and properly integrated with the posterior discussion regarding the Analysis of Key Parameters and Optimization.
- Some of the figures present evident mistakes. Many other can be improved. E.g. Fig. 5 should be made larger. Also, subfigures in the left column do not show the units for the Front-Rear Temperature. Difference. Figures 6 and 7 are confusing, since it is not clear which sets of curves correspond to the left Y axis and which to the right. Moreover Fig. 6 is too small, and the legend for Fig. 7 is splitted between two different pages. Maps in Figs. 9 and 10 are very small. The X axis label is wrong (compressuor?).
- There are many serious mistakes in the Notation section. The temperatures T_T, T_S, T_O are measured in what units? Kelvin is shortened to capital K. T_a seems to correspond to the standard atmospheric pressure? Whatever that quantity is, it is measured in Mpa. However the pressure unit that is 10^6 Pascal is shortened to MPa, with capital P. The U_T turbine wheel peripheral speed is measured in m/S; the unit that indicates seconds is a small s. The molar heat capacity (uc_p)_T is measured in Kelvin * Joules/(mole K)? Or is it kiloJoule/(mole Kelvin)? Scientific documents need to pay attention to this kind of details.
Minor comments:
- Mentions to the seminal work of Benson and Svetnicka without proper referencing in the bibliography.
- I believe that numbering the different sections of the manuscript could help to orient the reader. Is the numbering of sections compatible with the Energies format?
- Finally, I observe that the word “author” (in singular) is used through most of the work. However, there are five different people listed in the authors list. In the end, how many people contributed to the work?
Author Response
Dear Editors
First of all, we would like to thank the editors and the reviewers for sparing their precious time to review the manuscript and their comments are invaluable for us to further improve the quality of this work. A detailed response to the comments is listed below for your kind perusal.
Reviewer 1
Point 1: The presentation of the “Model equations” is not informative enough. The authors simply enumerate the equations which are the base of their work, without any explanation of how these equations are derived or how of these equations interact with each other. As a results, following the paper becomes unnecessarily difficult. At this point, I believe that a diagram (perhaps something as simple as a block/flow diagram) relating the equations used throughout the paper with the facility considered would have been very helpful. In summary, in a posterior version of this article, the presentation of the equations needs to be made clear and properly integrated with the posterior discussion regarding the Analysis of Key Parameters and Optimization.
Response1: Thank the reviewer a lot for pointing this out. In this paper, professor Gu hongzhong's JTK model from Shanghai jiao tong university [30] was used to calculate the total excess air coefficient by using the energy balance relationship of fuel. And then the total excess air coefficient is used to calculate the air intake flow rate of the engine, so as to obtain the pressure ratio. Because the model can only be applied to single stage turbocharge system, the model is optimized in this paper. JTK model was used to calculate the total turbocharger efficiency of the engine, and the two-stage turbocharge system was considered as a whole to calculate the total expansion ratio. Then the pressure ratio, expansion ratio and the energy distribution relationship of the turbocharge system were calculated according to the power balance equation of each stage of the turbocharge system. As suggested by the reviewer, a logic diagram was added in the revised manuscript.
Point 2: Some of the figures present evident mistakes. Many other can be improved. E.g. Fig. 5 should be made larger. Also, subfigures in the left column do not show the units for the Front-Rear Temperature. Difference. Figs 6 and 7 are confusing, since it is not clear which sets of curves correspond to the left Y axis and which to the right. Moreover Fig. 6 is too small, and the legend for Fig. 7 is splitted between two different pages. Maps in Figs. 9 and 10 are very small. The X axis label is wrong (compressuor?).
Response2: In the revised manuscript, the temperature unit (K) has been added Fig.6. The curves in Fig. 7 and Fig. 8 have been revised. Fig. 10 and Fig. 11 have been corrected.
Point 3:There are many serious mistakes in the Notation section. The temperatures T_T, T_S, T_O are measured in what units? Kelvin is shortened to capital K. T_a seems to correspond to the standard atmospheric pressure? Whatever that quantity is, it is measured in Mpa. However the pressure unit that is 10^6 Pascal is shortened to MPa, with capital P. The U_T turbine wheel peripheral speed is measured in m/S; the unit that indicates seconds is a small s. The molar heat capacity (uc_p)_T is measured in Kelvin * Joules/(mole K)? Or is it kiloJoule/(mole Kelvin)? Scientific documents need to pay attention to this kind of details.
Response3: Thank the reviewer for pointing out the mistake. All the units of temperature, air pressure and velocity have been revised.
Point 4: Mentions to the seminal work of Benson and Svetnicka without proper referencing in the bibliography.
Response 4: The citation has been added in the revised manuscript.
“R. S. Benson. Two—Stage Turbocharging of Diesel Engines: A Matching Procedure and Experimental Investigation. SAE 1974-09-12, 1974.”
Point 5: Finally, I observe that the word “author” (in singular) is used through most of the work. However, there are five different people listed in the authors list. In the end, how many people contributed to the work?
Response 5: Everyone contributed to this work. Sorry for the misunderstanding caused. In the revised manuscript, the “author” is changed to “authors”.
Finally, we really very appreciate the editors and reviewers for sparing precious time to review the paper. The editor and reviewer’s comments are invaluable for us to further improve the quality of this work.

Reviewer 2 Report
Why the coefficient 1.015 is used in equation (8) is not clear from the text. Figure 2 should define the relation between Total Supercharger Efficiency and Supercharger Raito or Expansion Ratio, but on the horizontal axis is the Efficiency of low-pressure stage compressor - in the text below the picture is a different symbol (designation) for the efficiency of high-pressure stage supercharger and the text is very confusing for reader.
Also, Figure 3 defines the effect of changing the low-pressure stage supercharger efficiency, but the horizontal axis shows the efficiency of the low-pressure stage compressor. So, the defined front-rear temperature difference has a different designation in the figure and in the text below the figure. In general, the text of the paper in some parts does not correspond to the mentioned graphic outputs. The graphic outputs themselves are harder to read, there is no obvious difference between High-pressure and Low-pressure stages.
As a result of these inaccuracies, the overall clarity of the text is low. I recommend choosing another type of graphical output and above all to correct inconsistencies in texts that describe graphical output.
Author Response
Dear Editors
First of all, we would like to thank the editors and the reviewers for sparing their precious time to review the manuscript and their comments are invaluable for us to further improve the quality of this work. A detailed response to the comments is listed below for your kind perusal.
Point 1: Why the coefficient 1.015 is used in equation (8) is not clear from the text. Figure 2 should define the relation between Total Supercharger Efficiency and Supercharger Raito or Expansion Ratio, but on the horizontal axis is the Efficiency of low-pressure stage compressor - in the text below the picture is a different symbol (designation) for the efficiency of high-pressure stage supercharger and the text is very confusing for reader.
Response 1: we are sorry for the misunderstanding caused. In the Eq.8, the coefficient 1.015 is used taking into account the mass flow proportion relation between exhaust and intake, which refers to Professor Gu hongzhong's JTK model of Shanghai jiao tong university [30]. In the revised manuscript Figure 3 has been corrected, and the X-axis has been changed to the total turbocharger efficiency.
Point 2: Figure 3 defines the effect of changing the low-pressure stage supercharger efficiency, but the horizontal axis shows the efficiency of the low-pressure stage compressor. So, the defined front-rear temperature difference has a different designation in the figure and in the text below the figure. In general, the text of the paper in some parts does not correspond to the mentioned graphic outputs. The graphic outputs themselves are harder to read, there is no obvious difference between High-pressure and Low-pressure stages.
Response2: We are sorry for the misunderstanding caused. The efficiency of low-pressure stage turbocharger and high-pressure stage turbocharger have been unified and the correction was made in the revised manuscript.
Point 3: As a result of these inaccuracies, the overall clarity of the text is low. I recommend choosing another type of graphical output and above all to correct inconsistencies in texts that describe graphical output.
Response3: We thank the reviewer for pointing these out. We have tried to revise all the inconsistencies caused throughout the text.
Finally, we really very appreciate the editors and reviewers for sparing precious time to review the paper. The editor and reviewer’s comments are invaluable for us to further improve the quality of this work.

Reviewer 3 Report
The document presents a model of the supercharging system of an automotive engine for heavy industrial use. The description of the work in the abstract should be improved. The formulas applied to limit the system optimizing the use of the two compressors seems correct. However this optimization is done for not all the operating points of the compressors. The authors must show the motor performance curve in each compressor operation map to observe if said optimization is not physically limited by phenomena such as surge or choke or inlet manifold inlet temperature. In the bibliography, there are several contributions on bi-turbo or bi-compressor models. These models take in account the flow direction and the outlet pressure of both. The operation must be coordinated by means of control valves that must be adapted to the model to know if the operation point with the two turbos or two compressors is possible. The operation maps of both compressors must be show. The bibliography is rich in this topic one expample of them is the work published by Skopil analysing two compressor in parallel.
. in formula two the subscript "as" is missing
Author Response
Dear Editors
First of all, we would like to thank the editors and the reviewers for sparing their precious time to review the manuscript and their comments are invaluable for us to further improve the quality of this work. A detailed response to the comments is listed below for your kind perusal.
Point 1: The document presents a model of the supercharging system of an automotive engine for heavy industrial use. The description of the work in the abstract should be improved. The formulas applied to limit the system optimizing the use of the two compressors seems correct. However this optimization is done for not all the operating points of the compressors. The authors must show the motor performance curve in each compressor operation map to observe if said optimization is not physically limited by phenomena such as surge or choke or inlet manifold inlet temperature. In the bibliography, there are several contributions on bi-turbo or bi-compressor models. These models take in account the flow direction and the outlet pressure of both. The operation must be coordinated by means of control valves that must be adapted to the model to know if the operation point with the two turbos or two compressors is possible. The operation maps of both compressors must be show. The bibliography is rich in this topic one expample of them is the work published by Skopil analysing two compressor in parallel.
Response1: Thank you for your comments. The abstract has been modified accordingly. According to the results of model calculation, although the pressure ratio and expansion ratio under the optimal distribution condition of two-stage turbocharging energy can be obtained, the matching relationship of turbocharger efficiency, intake flow rate, pressure ratio and expansion ratio calculated by the selected turbocharger models should be satisfied. The closer the matching characteristic is to the calculation result, the lower the energy consumption of two-stage turbocharging system is. If the optimal value is calculated away from the model, the turbocharger may be selected to work in the surge or blocking area. Therefore, the constraint conditions of model adaptability are added in this paper: the turbocharger efficiency of all stages >0.42, the turbocharging ratio and expansion ratio of all stages >1, the intake manifold temperature <330K, and the turbine front temperature <923K, to increase the applicability of the model. This paper focuses on the distribution constraint equation of the two-stage turbocharger pressure ratio and expansion ratio for the optimal energy consumption of two-stage turbocharger system. It can be realized by the matching of two-stage turbocharger, or can be realized by adding bypass valves. Due to the large amount of content, it will be discussed in detail in the next article submitted soon. We thank the reviewer for giving the invaluable comments.
Point 2: in formula two the subscript "as" is missing
Response2: Formula 2 is correct. Because “"is the excess air coefficient, “"Is the total excess air coefficient,“"is the scavenging coefficient. However,=*
Finally, we really very appreciate the editors and reviewers for sparing precious time to review the paper. The editor and reviewer’s comments are invaluable for us to further improve the quality of this work.

Round 2
Reviewer 1 Report
I have read carefully the revised manuscript entitled 'Establishment of Two-stage Supercharging System Model and Analysis on Influence Rules of Key Parameters', under consideration for publication in the Energies Journal.
I feel that my major concerns have been satisfactorily addressed, and therefore I recommend the ACCEPTANCE of the manuscript in its current form.